# Improved Interpretability Without Performance Reduction in a Sepsis Prediction Risk Score

**DOI:** 10.3390/diagnostics15030307

**Published:** 2025-01-28

**Authors:** Adam Kotter, Samir Abdelrahman, Yi-Ki Jacob Wan, Karl Madaras-Kelly, Keaton L. Morgan, Chin Fung Kelvin Kan, Guilherme Del Fiol

**Affiliations:** 1Department of Biomedical Informatics, University of Utah, Salt Lake City, UT 84108, USA; samir.abdelrahman@utah.edu (S.A.); jacob.wan@utah.edu (Y.-K.J.W.); keaton.morgan@hsc.utah.edu (K.L.M.); guilherme.delfiol@utah.edu (G.D.F.); 2College of Pharmacy, Idaho State University, Meridian, ID 83209, USA; karlmadaraskelly@isu.edu; 3Department of Emergency Medicine, University of Utah, Salt Lake City, UT 84112, USA; 4Department of Anesthesiology, University of Utah, Salt Lake City, UT 84112, USA; kelvin.kan@hsc.utah.edu

**Keywords:** interpretability, explainability, sepsis prediction, integer risk scores, logistic regression, temporal reasoning, clinical decision support

## Abstract

**Objective**: Sepsis is a life-threatening response to infection and a major cause of hospital mortality. Machine learning (ML) models have demonstrated better sepsis prediction performance than integer risk scores but are less widely used in clinical settings, in part due to lower interpretability. This study aimed to improve the interpretability of an ML-based model without reducing its performance in non-ICU sepsis prediction. **Methods**: A logistic regression model was trained to predict sepsis onset and then converted into a more interpretable integer point system, STEWS, using its regression coefficients. We compared STEWS with the logistic regression model using PPV at 90% sensitivity. **Results**: STEWS was significantly equivalent to logistic regression using the two one-sided tests procedure (0.051 vs. 0.051; *p* = 0.004). **Conclusions**: STEWS demonstrated equivalent performance to a comparable logistic regression model for non-ICU sepsis prediction, suggesting that converting ML models into more interpretable forms does not necessarily reduce predictive power.

## 1. Introduction

Sepsis is defined as a “life-threatening organ dysfunction caused by a dysregulated host response to infection” [1]. Sepsis is a major cause of hospitalization and may contribute to over a third of all hospital deaths [2,3]. In patients with severe forms of sepsis, the risk of in-hospital mortality increases with each one-hour delay in administration of antibiotics by as much as 7.6% per hour [4,5]. The CDC has recognized that initiatives “focused on recognition and early management of sepsis have been associated with reductions in in-hospital mortality” [6]. Therefore, earlier detection of sepsis is vital for patient health.

Despite the importance and availability of scoring systems such as SOFA [7] and qSOFA [8] for the prediction and detection of sepsis outside of intensive care unit (ICU) settings, systematic reviews have shown that the majority of machine learning (ML)-based sepsis prediction research has focused on ICUs [9,10,11]. ICUs differ from hospital wards in that more frequent, consistent, and comprehensive patient assessments are performed. In addition, ICU patients have a significantly higher sepsis incidence and mortality rate than non-ICU patients [3]. Thus, sepsis-prediction solutions for non-ICU patients need to be tailored to a non-ICU setting, in particular by addressing data sparsity, in terms of both predictors and prediction events.

Many different solutions have been attempted for early detection or prediction of sepsis, such as integer point-based early warning systems (EWSs) or ML methods. ML approaches have consistently outperformed EWSs in terms of sensitivity and area under the receiver operating characteristic curve (AUROC) [9]. Nevertheless, implementation challenges such as a low level of transparency [12], installation and engineering [13], regulatory requirements, and changing clinician behavior [14] prevent wider adoption of these ML models. Instead, integer-based EWSs have been much more widely adopted in spite of their lower performance, potentially due to their greater interpretability and preexisting familiarity to clinicians [12,15]. Because the relative contribution of each variable to the risk prediction in an integer-based score can be easily obtained and displayed to users without any additional arithmetic, integer point-based scores are naturally transparent and interpretable. Incorporating this interpretability into ML algorithms could lead to wider implementation of predictive ML algorithms in clinical settings [12].

Unfortunately, a tradeoff between interpretability and performance in ML algorithms is thought to occur. While this tradeoff is complex and varies across both model types and applications, the highest-performing ML models tend to have the lowest interpretability, as increased complexity in model design tends to simultaneously increase model performance and decrease model interpretability [16,17]. Studies are needed to achieve ML models that are both high-performing and easily interpretable for clinical implementation.

The overall goal of this study was to produce a more interpretable integer EWS (STEWS, Sepsis Temporal EWS) that predicts sepsis in non-ICU patients from a less interpretable ML model without compromising performance. We hypothesized that STEWS would have equivalent performance to a simple, comparable ML model (logistic regression).

## 2. Methods

### 2.1. Overview

We aimed to compare the sepsis-predicting performance of a class of interpretable integer-based EWS scores (STEWS scores) derived from logistic regression over temporal features with a less interpretable baseline of the initial logistic regression in non-ICU settings. The primary hypothesis was that STEWS would have equivalent performance to logistic regression over temporal features in terms of positive predictive value (PPV) at a fixed sensitivity of 90%. For reference, PPV is defined as the ratio between correctly identified positive cases and all cases identified as positive, and sensitivity is defined as the ratio between correctly identified positive cases and all cases that were actually positive. Secondary hypotheses included the following: (i) STEWS has higher PPV with longer versus shorter observation windows, and (ii) STEWS outperforms commonly used EWS scores.

### 2.2. Dataset

We trained and tested STEWS using data from University of Utah Health (UHealth). The study was approved under Institutional Review Board (IRB) protocol number IRB_00152146. The dataset included hospital admissions of adults to non-ICU settings (i.e., the wards where model use would be relevant) between 1 June 2014 and 2 November 2022 with a duration of at least 36 h. The following admissions were excluded: (1) admissions to non-medical–surgical wards, such as labor and delivery and oncology, and (2) admissions where the patient was discharged or transferred to an ICU within 36 h of the admission (Figure 1). Admissions where the patient was first admitted to the emergency department and subsequently transferred to a ward were included. The 36 h minimum sepsis-free admission period was selected to increase the likelihood that at least one feature measurement was present in each observation interval used by a STEWS score prior to sepsis prediction.

Of an initial total of 315,925 hospital admissions, 51,712 admissions met the inclusion criteria (Figure 1). The basic unit of analysis used in this study was the hour. The included admissions lasted for a combined total of 5,745,913 h, of which only 1,812,273 h (~31.5%) contained at least one patient measurement relevant to the prediction models (see Table 1 for patient demographic data).

### 2.3. Sepsis Definition

We defined sepsis according to the Sepsis-3 criteria: (1) identification of suspected infection based on orders for blood cultures and antibiotics and (2) a clinical criterion for life-threatening organ dysfunction [1]. We operationalized suspicion of infection as concurrent orders for blood cultures and initiation of intravenous antibiotics administered for four out of seven consecutive days (Figure 2). The inclusion of the intravenous antibiotics criterion alongside the blood culture criterion limited the potential for blood culture sampling independent of suspected infection, which might have produced false positives in the gold standard. Following the procedure of [18], the orders were considered concurrent if either of the following conditions were met: the blood culture was ordered first AND the antibiotic was ordered within 72 h of the blood culture, OR the antibiotic was ordered first AND the blood culture was ordered within 24 h of the antibiotic. We operationalized life-threatening organ dysfunction as the presence of any hospital discharge ICD9 or ICD10 codes from the Martin, Angus, CMS, or Explicit code sets [19]. Patients were classified as having sepsis if they met both the suspicion of infection criterion AND the presence of life-threatening organ dysfunction criterion. Time of sepsis onset was defined as the first time either of the orders indicating suspicion of infection was placed (i.e., the order for blood culture or the order for intravenous antibiotic administration) (Figure 2). This sepsis definition operationalization is consistent with the highest-mortality cohort in [20]’s investigation of suspicion of infection criteria (which did not significantly lower prediction accuracy) and the broadest selection of life-threatening organ dysfunction codes from [19]’s comparison of these codes.

We classified each hour of each patient hospitalization as either sepsis-positive or sepsis-negative. The STEWS model was then trained to predict this outcome. An hour was labeled as sepsis-positive if sepsis onset (according to the study criteria) occurred from the following hour to a set number of hours after the hour of interest. For example, if the number of hours for prediction was set as 6, and sepsis onset for a given patient occurred at hour 16 of admission, then hours 10 through 15 would be labeled as sepsis-positive. The hour of onset itself and all of the following hours were not considered in this study because the treatment of sepsis inherent in the study’s sepsis definition would significantly change patient care (and thus model performance). Different numbers of hours for prediction were tested because different prediction window lengths may have different utility in different situations.

### 2.4. Clinical Variables

We used clinical variables commonly measured in general wards and frequently used in previous studies [10] as inputs for the STEWS scores. These included the following: heart rate, respiratory rate, diastolic blood pressure, systolic blood pressure, oxygen saturation, temperature (in degrees Celsius), age, and administrative sex. Each of these variables is either a direct indicator of patient health or a demographic factor (e.g., age [21] and sex [22]) associated with sepsis risk. The STEWS scores did not use any laboratory values as inputs, which offers an advantage over models that rely on those predictors due to time delays between ordering laboratory tests and receiving the results. Other integer EWS scores included for comparison used laboratory values and other measurements, including platelet count, white blood cell count, bilirubin, creatinine, fraction of inspired oxygen (FiO_2_), and Glasgow Coma Scale (GCS) score. To account for inconsistent measurement intervals, each raw measurement was considered to occur at the beginning of the hour in which it was taken. If multiple measurements of the same variable were taken in the same hour, the average of those measurements was used. Age and administrative sex were assumed to be constant across the entire admission period.

### 2.5. Temporal Method

Temporal methods are processes by which timestamped data outside the present moment are used in algorithms. Temporal methods can potentially benefit clinical prediction by using timeframes broader than the present moment to make predictions, enabling utilization of more information and context to improve model performance, although the exact benefits of including more information depend on the length of prediction windows and the purpose of the prediction [23]. While there are no standards or consensus for how far into the past temporal methods can reach for data (i.e., observation window length), temporal methods generally focus on time periods when the modeled signal is present [24], yet also far enough back to gather low-frequency data [25]. Typical observation window lengths for ICU sepsis prediction are 6–12 h [25,26]. Observation windows for mixed ICU and non-ICU sepsis prediction have reached up to five days [27]. To maximize STEWS’ predictive performance, we used a temporal method to abstract our data. Because vital sign measurements of non-ICU patients may be infrequent, a temporal method that is robust to low-frequency observations was used. For every hour in the dataset, we divided the hours prior to the hour of interest into a set of four equal-length intervals, where the length of the interval was a parameter that varied between experiments (Figure 3). If the hour of interest occurred within four times the interval length from admission, then the intervals were reduced to each cover a quarter of the hours between admission and the hour of interest. Two additional intervals were also used: the interval from 168 h prior to admission up to admission and the interval from admission to the hour of interest. The interval prior to admission included data gathered from admissions to the emergency department immediately preceding the considered admission.

Over each interval, summary statistics (minimum, median, maximum, mean, and standard deviation) were calculated for each vital sign (Figure 3). Multiple experiments were conducted varying the maximum interval length to determine optimal model settings. All clinical variables used as model inputs (excluding age and administrative sex, which were assumed to be constant) were transformed with this temporal method before use in the model.

### 2.6. Development of STEWS Scores

For each training iteration of a STEWS score, we first created a 70–30 train–test split by patient. The hours from a given patient were either all training or all testing, not both. Next, we randomly downsampled sepsis-negative hours in each dataset such that the training set had a 50% sepsis incidence by hour and the test set had a 5% sepsis incidence by hour. A 50% sepsis incidence was chosen to mitigate the negative impact of class imbalance on the STEWS scores’ performance, and the 5% incidence value was chosen in consultation with expert clinicians to approximate hospital sepsis rates. Each datapoint in these sets represented one hour of a patient admission temporally abstracted to contain information about the preceding hours of that same admission. Hours with missing features were imputed using carry-forward imputation where possible or using physiologically normal data otherwise.

The first step in developing each iteration of the STEWS scores was to narrow the list of input features (including temporal features) by performing univariate and then multivariate feature selection. We used the two-sample, two-tailed Kolmogorov–Smirnov test with α = 0.05 for univariate selection to determine whether feature values were significantly different between sepsis-positive and sepsis-negative hours for each model input feature. Features without any significant difference between sepsis-positive and sepsis-negative hours were removed. Multivariate feature selection was performed by calculating the variance inflation factor (VIF) for each feature. Features with a calculated VIF strictly greater than 10 were removed to limit multicollinearity. As such, each STEWS score, though developed with the same method, could have different input features based on the subset of data on which it was trained.

The remainder of STEWS development followed the general flow presented by the Framingham Study risk score [28] to generate regression weights and convert those weights to integers. We discretized each of the selected model inputs (except administrative sex) by performing one-hot encoding of each feature by quintiles. The testing data was discretized using the quintiles identified in the training data. We trained a logistic regression model on the discretized model inputs using a hyperparameter tuning search across inverse regularization strength *C* (values 0.1, 1, 10, 100, and 1000) and L1 ratio (values 0, 0.2, 0.4, and 0.6). We used a relatively small search space to reduce computational complexity and kept L1 ratios at or below 0.6 to avoid feature weights being set to zero.

After training the logistic regression model for an iteration of STEWS development, we converted the logistic regression model into an integer point-based system. We used the minimum absolute value of non-zero model weights as a conversion factor from regression units to points. The point value of each discretized feature value was calculated as its regression weight divided by the conversion factor and rounded to an integer. The point values were then scaled linearly and re-rounded so that no feature could have a maximum point value greater than 5 or a minimum point value less than −5. More significant features have a greater maximum absolute point value, to the maximum of 5.

This conversion to an integer point-based system makes STEWS structurally similar to the integer-based EWSs commonly used by clinicians. Crucially, this allows clinicians to more easily determine the relative contribution of each STEWS variable to the risk predicted by STEWS with as little arithmetic as possible. Because the underlying logistic regression model requires multiplication of coefficients by regressors to determine contribution to risk prediction, this makes STEWS more intuitively interpretable than its underlying logistic regression model.

The final STEWS score is calculated as the sum of point values for all features. Since each quintile of each input feature has an associated point value, each feature contributes one of up to five different point values to the total STEWS score. The maximum potential range of point values is from −5 times the number of features to +5 times the number of features. The point threshold at which STEWS gives a positive result is set such that model sensitivity on the test set is as close as possible to 0.90, a value determined in consultation with expert clinicians to avoid missing sepsis cases. The STEWS pseudocode algorithm is available upon request. Because a different model was created with each iteration of STEWS, there is no single set of point values for STEWS scores.

### 2.7. Statistical Analyses

All experiments used PPV as the primary outcome at a fixed sensitivity of 90%. Secondary outcomes included AUROC and F1-score. Bonferroni-corrected significance levels were used for hypotheses with multiple statistical tests where the significance of any one test independent of the other tests would be meaningful [29]. Otherwise, the traditional α = 0.05 significance level was used. All tests were conducted on subsets of the UHealth data complementary to their respective training subsets and where 5% of patient visits included a sepsis event.

### 2.8. Secondary Hypothesis 1—STEWS Has Higher PPV with Longer Versus Shorter Observation Windows

We trained *n* = 60 iterations of STEWS scores using a 3 × 4 factorial design with 3 prediction window lengths (6, 8, and 12 h) and four observation window lengths (9, 12, 15, and 18 h). Five STEWS iterations were assigned to each of the twelve combinations of prediction and observation window length, and each iteration was trained on a different random subset of the UHealth dataset. We performed a regression analysis with each observation window, each prediction window, and each combination of observation and prediction windows as a one-hot encoded regressor to predict PPV. This provided three coefficients for observation windows, four coefficients for prediction windows, and twelve coefficients for interaction effects, as well as a constant. We used F-tests to test for equivalence of the observation length coefficients, equivalence of the prediction length coefficients, and equivalence of the interaction effects. The results of these tests informed our choice of optimal observation interval and prediction interval lengths in the remaining experiments. Because significance in any of these three F-tests would indicate a meaningful relationship between window length and PPV, we used a Bonferroni-corrected significance level of α = 0.017.

### 2.9. Secondary Hypothesis 2—STEWS Outperforms Commonly Used EWS Scores

Using the optimal observation and prediction interval lengths identified in Hypothesis 1, we trained *n* = 96 iterations of STEWS. On the same test sets we used for the STEWS scores, we also tested the following diagnostic EWS scores: NEWS, SIRS, and PRESEP. We chose diagnostic EWS scores rather than prognostic scores because our prediction task more closely matched identification of patients that may have sepsis (diagnosis) rather than stratification of patients by sepsis risk (prognosis), and meta-analyses and individual studies have suggested that diagnostic scores tend to outperform prognostic scores for diagnostic tasks [30,31,32,33]. We used the Friedman chi-square test to test for a significant difference between scores, and if this overall difference was significant, we used the Nemenyi post hoc test for pairwise comparisons. Because a significant difference between a version of STEWS and any of the three other EWS scores would indicate that STEWS’ performance was not equivalent to the performance of the other EWS scores, we used a Bonferroni-corrected significance level of α = 0.017.

### 2.10. Primary Hypothesis—STEWS Has Equivalent Performance to Temporal Logistic Regression

The performance of the less interpretable baseline temporal logistic regression was also obtained during the experiments of Secondary Hypothesis 2. We performed two one-sided tests with the Wilcoxon signed-rank test (Wilcoxon TOST) and equivalence bounds of ±0.001 to compare the mean PPVs of STEWS and its equivalent logistic regression models. The two one-sided tests evaluated the null hypotheses that (1) STEWS’ PPV performance is lower than logistic regression’s performance by at least 0.001 and that (2) STEWS’ PPV performance is greater than logistic regression’s performance by at least 0.001. Rejection of both of these null hypotheses indicates that STEWS’ performance is significantly within 0.001 of logistic regression’s performance and that the two are thus functionally equivalent. Because significance on just one test would not be sufficient to establish equivalence, we did not use a Bonferroni correction for this hypothesis.

## 3. Results

Of the 51,712 admitted patients who met the inclusion criteria, 301 patients (0.6%) met the sepsis definition, 305 patients (0.6%) expired, and 2832 patients (5.5%) were transferred to the ICU (Table 1).

### 3.1. Secondary Hypothesis 1—STEWS Has Higher PPV with Longer Versus Shorter Observation Windows

At a sensitivity of 90%, the PPVs for the different combinations of observation windows and prediction windows varied from 0.054 to 0.058 (Table 2), but the differences (including interaction effects) were not significant (*p* = 0.321). As such, the subsequent experiments used the simplest approach with the shortest time intervals, namely, a 9 h observation interval and 6 h prediction window, which resulted in a PPV of 0.058 (0.056–0.061, 95% CI; Table 2).

### 3.2. Secondary Hypothesis 2—STEWS Outperforms Commonly Used EWS Scores

NEWS and SIRS had significantly higher PPVs than STEWS (0.053 vs. 0.067 vs. 0.051; *p* < 0.001). PRESEP and STEWS were not significantly different (0.051 vs. 0.051; *p* = 0.563) (Table 3).

### 3.3. Primary Hypothesis—STEWS Has Equivalent Performance to Temporal Logistic Regression

STEWS and its less interpretable underlying logistic regression model had significantly equivalent PPV (0.051 vs. 0.051, equivalence bounds ±0.001, *p* = 0.004; Table 4).

## 4. Discussion

We used commonly measured vital signs and an interval summary statistics temporal method to develop STEWS, a class of interpretable integer risk scores that predict sepsis onset in non-ICU hospital settings using temporally abstracted patient data. Our purpose was to produce a more interpretable integer risk score from a less interpretable ML model without compromising the ML model’s performance. In terms of PPV at a fixed sensitivity of 90%, STEWS had significantly equivalent performance to the comparable logistic regression models with equivalence bounds of ±0.001. This suggests that converting ML models to more interpretable, clinician-friendly forms does not necessarily reduce predictive power, although this finding stands in contrast to previous work suggesting that more complex models perform better than simpler models [34]. Developers of clinical decision support (CDS) systems may wish to investigate simplifying their predictive models to enhance clinician usability without necessarily decreasing predictive performance.

We found that STEWS scores were significantly outperformed by SIRS and NEWS, two commonly used non-temporal EWS scores. Widespread use of SIRS and NEWS could have had a role in determining clinician behavior, which is incorporated in sepsis prediction labels (e.g., blood culture orders and antibiotic administration), and thus may have contributed to SIRS and NEWS outperforming STEWS. The small pool of sepsis-positive patient admissions eligible for use in score development in the study dataset (*n* = 301) may also have contributed to the low performance of STEWS scores.

Finally, we found that the length of the time intervals over which data are temporally abstracted or over which predictions are made did not significantly affect the performance of STEWS scores in terms of PPV at a fixed sensitivity of 90%. This result aligns with previous findings that increasing the number of observations used to make predictions does not improve predictive performance after a certain point [35] but contrasts with prior work suggesting that shorter prediction horizons and longer observation windows may improve predictive performance [23]. Our results suggest that CDS developers who produce temporally aware predictive models may wish to use shorter data-gathering windows to focus on times more likely to contain relevant clinical signals and to reduce the computational complexity of temporal analysis. Prediction window selection may benefit from focusing less on optimizing predictive performance and focusing more on the actionability of predictions.

### Limitations and Future Work

A major limitation of this work was the small number of sepsis-positive samples available for training the model to develop STEWS scores. In order to have sufficiently long temporal intervals for STEWS to make predictions, we did not consider any sepsis-positive cases with onset within the first 36 h of admission, which excluded the majority of sepsis-positive cases from our training sets. Additionally, this observation window length prevents STEWS scores from being used to monitor newly admitted patients, which limits the utility of STEWS scores. Future work should focus on acquiring larger datasets and developing temporal risk scores that use shorter temporal intervals. Using multi-site data in future work would grant the benefit of larger datasets as well as more generalizable findings.

Another direction for future work would be to use ML methods beyond logistic regression. Simplifying different ML models, including non-linear models, could generalize our findings about the tradeoffs between model complexity and model performance. Using clinical conditions apart from sepsis could also generalize our findings.

Finally, although most hospitals in high-resource countries such as the USA have automated data collection for the predictors used in STEWS scores, other settings with fewer resources still rely on manual data entry. This imposes data availability and timeliness challenges in the implementation of automated scoring systems.

## 5. Conclusions

STEWS demonstrated equivalent performance to a comparable but less interpretable logistic regression model at the task of non-ICU sepsis prediction, although it did not outperform commonly used integer risk scores. The foundational advancement of this work is the conclusion that converting ML models into more interpretable, clinician-friendly forms does not necessarily reduce predictive power.

## Figures and Tables

**Figure 1 diagnostics-15-00307-f001:**
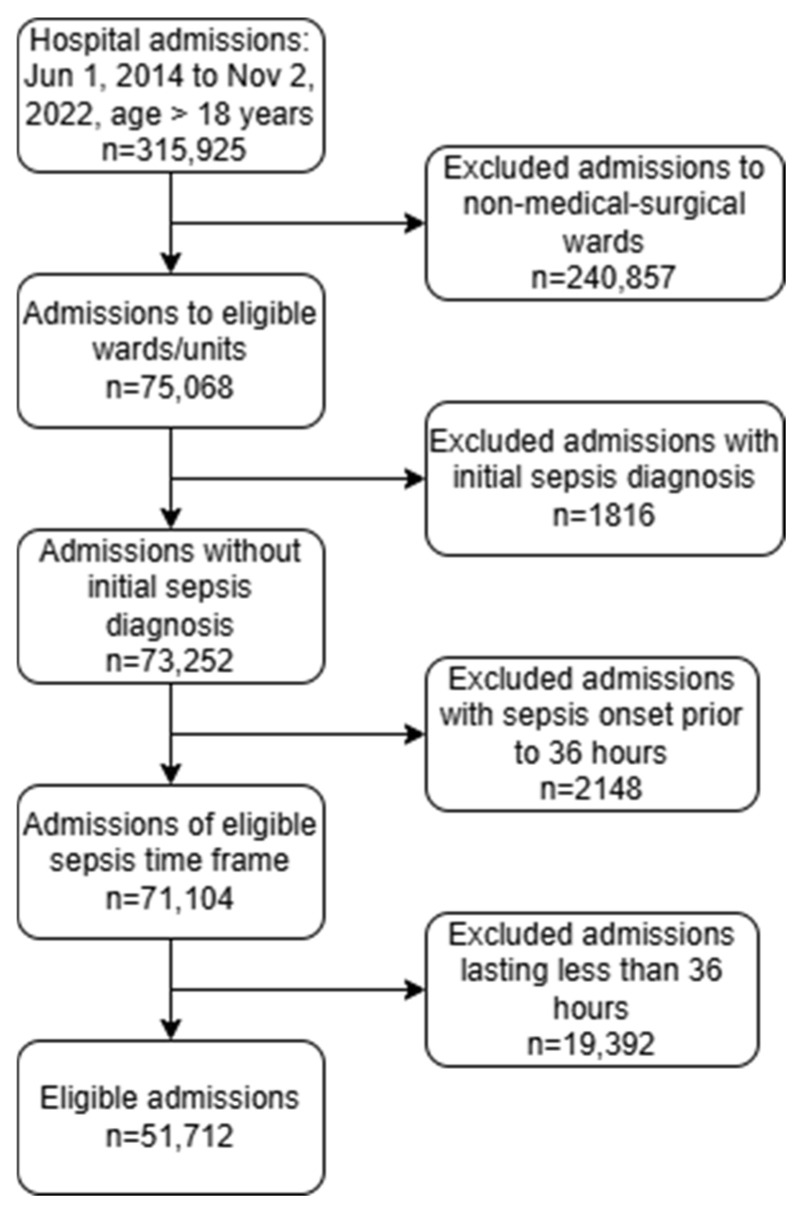
Study data flow.

**Figure 2 diagnostics-15-00307-f002:**
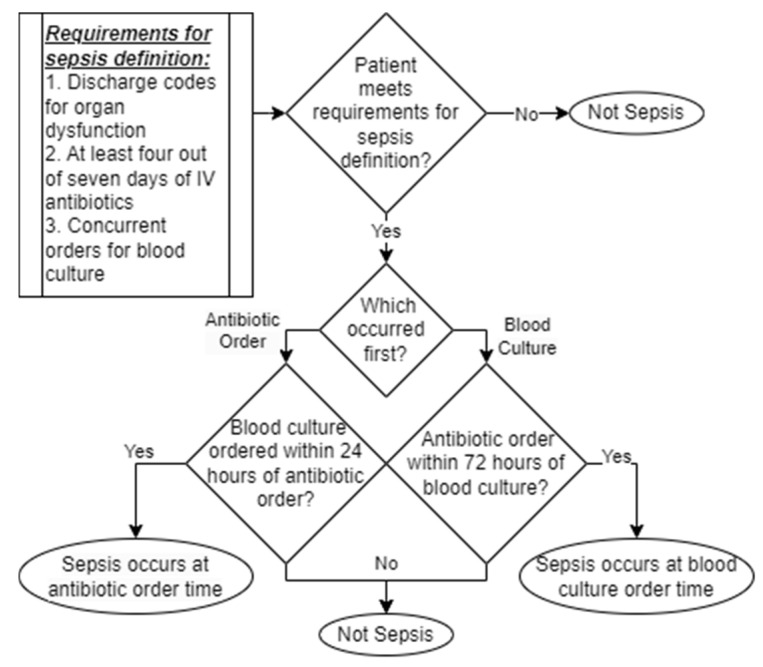
Sepsis definition implementation.

**Figure 3 diagnostics-15-00307-f003:**
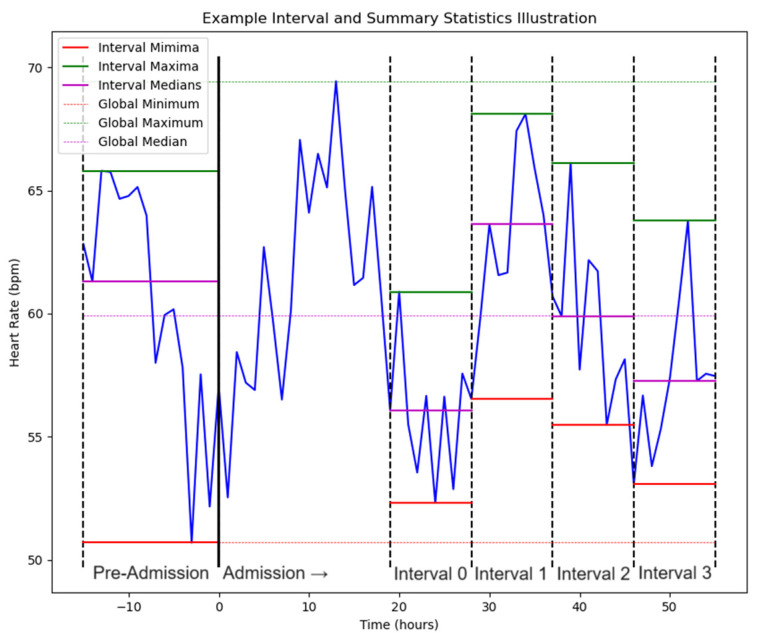
Illustration of example intervals and summary statistics with minima, maxima, and medians illustrated over each interval. For clarity, the standard deviation and mean are not represented in the graph. The admission interval covers all time after time 0, as indicated by the arrow.

**Table 1 diagnostics-15-00307-t001:** Patient demographic data.

Category	Count (%)
Administrative Sex	
Male	26,914 (52.0%)
Female	24,797 (48.0%)
Other/Unknown	1 (0.0%)
Admission Unit	
Emergency Department	34,340 (66.4%)
Cardiovascular Medical Unit *	4695 (9.1%)
Neurological Acute Care	4129 (8.0%)
Surgical Transplant Unit *	4039 (7.8%)
Acute Internal Medicine *	4021 (7.8%)
Other Acute Care *	488 (0.9%)
Race	
White or Caucasian	41,679 (80.6%)
Black or African American	1283 (2.5%)
American Indian and Alaska Native	1267 (2.5%)
Asian	840 (1.6%)
Other Pacific Islander	836 (1.6%)
Unknown	687 (1.3%)
Other	5120 (9.9%)
Ethnicity	
Not Hispanic/Latino	44,760 (86.6%)
Hispanic/Latino	5604 (10.8%)
Unknown	1347 (2.6%)
Native Hawaiian/Other Pacific Islander	1 (0.0%)
Patient Endpoint	
ICU Transfer	2832 (5.5%)
Expired	305 (0.6%)
Sepsis	301 (0.6%)
Discharged, Other Transfer, Other	48,274 (93.4%)

* comprises at least two units with similar names and purposes.

**Table 2 diagnostics-15-00307-t002:** Mean PPVs by observation interval length (with 95% confidence intervals).

Observation Length	6 h Prediction	8 h Prediction	12 h Prediction
9 h Interval	0.058 (0.056–0.061)	0.058 (0.056–0.060)	0.056 (0.054–0.058)
12 h Interval	0.056 (0.053–0.058)	0.057 (0.055–0.059)	0.058 (0.055–0.060)
15 h Interval	0.057 (0.053–0.060)	0.057 (0.054–0.059)	0.054 (0.053–0.056)
18 h Interval	0.057 (0.054–0.060)	0.056 (0.054–0.058)	0.055 (0.053–0.056)

**Table 3 diagnostics-15-00307-t003:** Mean PPVs by EWS (with 95% confidence intervals).

Score	PPV at 90% Sensitivity	*p*-Value (Different from STEWS, Nemenyi Post Hoc)
STEWS	0.051 (0.050–0.051)	-
NEWS	0.053 (0.053–0.054)	*p* < 0.001
SIRS	0.067 (0.064–0.069)	*p* < 0.001
PRESEP	0.051 (0.050–0.051)	*p* = 0.563
*p*-value (Friedman chi-square test)	*p* < 0.001	

**Table 4 diagnostics-15-00307-t004:** Mean PPVs and *p*-values of STEWS versus logistic regression (with 95% confidence intervals).

STEWS (PPV)	0.051 (0.050–0.051)
Logistic Regression (PPV)	0.051 (0.050–0.051)
*p*-value (H_0_: STEWS ≥ LogReg + 0.001)	*p* = 0.004
*p*-value (H_0_: STEWS ≤ LogReg − 0.001)	*p* < 0.001

## Data Availability

Restrictions apply to the availability of these data. Data were obtained from University of Utah Health with permission.

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
