# Peer review of "Improved Interpretability Without Performance Reduction in a Sepsis Prediction Risk Score"

_diagnostics, 2025, doi:10.3390/diagnostics15030307_

Round 1
Reviewer 1 Report
Comments and Suggestions for Authors
Thank you very much for the opportunity to review this manuscript.
The presentation of the detection of sepsis as being primarily or predominantly related to intensive care units is misleading in this context. It is not without reason that the Sepsis 3 initiative has focused on the task of detecting patients who require observation and thus first admission to an ICU, for example with the help of the qSOFA and the SOFA score.
Unfortunately, the authors make little mention of the fact that the key point of using IT-based methodologies such as ML depends on achieving high quality data as quickly as possible.
Here it is certainly of crucial importance to point out in the introduction that the detection of sepsis based on laboratory chemistry or other more complex diagnostics depends primarily on the results and the resulting time delay.
There are considerable difficulties here, even in high-performance medical societies such as the USA.
For what reason were obstetrics (puperal sepsis) or oncology (high-risk sepsis patients) patients excluded?
Perhaps the authors also named the SOFA score used so that it is clear that numerous laboratory parameters were included in the identification of sepsis.
Did the authors consider that there may have been operationalized blood culture sampling independent of suspected sepsis? (e.g. sterile new insertion of invasive catheters), this could possibly have the consequence that fewer than just under 300 cases of sepsis were observed.
Unfortunately, the authors are somewhat neglectful in the classification of their work. Very few hospitals use ML systems at all, and many hospitals, especially in middle or low income countries, have not even implemented any form of digitalization. And even those that do have such a PDMS infrastructure rely to a large extent on manual data entry. In addition to correct data entry, this is one of the sometimes completely unsophisticated implementation problems of all systems. In this respect, the development of STEWS is academically exciting, but how this should help in clinical practice and what steps should be taken to ensure data quality and what clinical consequences should arise from a positive risk score remain unclear.
Author Response
Summary
Thank you for taking the time to review this manuscript. Please find the detailed responses below and the corresponding revisions highlighted and in track changes in the re-submitted file.
Point-by-point response to Comments and Suggestions for Authors
Comments 1: Thank you very much for the opportunity to review this manuscript.
The presentation of the detection of sepsis as being primarily or predominantly related to intensive care units is misleading in this context. It is not without reason that the Sepsis 3 initiative has focused on the task of detecting patients who require observation and thus first admission to an ICU, for example with the help of the qSOFA and the SOFA score.
Response 1: Thank you for your concern and for pointing out the importance of monitoring patients who may require admission to an ICU. We have modified a line as follows in the Introduction section to address the importance of sepsis detection outside of ICUs (paragraph 2 of page 1, lines 32-35).
Despite the importance and availability of scoring systems such as SOFA [7] and qSOFA [8] for prediction and detection of sepsis outside of intensive care unit (ICU) settings, systematic reviews have shown that the majority of machine learning-based (ML) sepsis prediction research has focused on ICUs [9-11].
Comments 2: Unfortunately, the authors make little mention of the fact that the key point of using IT-based methodologies such as ML depends on achieving high quality data as quickly as possible.
Response 2: Thank you for pointing this out. We have added a line as follows in the Limitations section (paragraph 4 of page 10, lines 371-373) addressing the problem of lengthy observation windows delaying data acquisition and prediction-making.
Additionally, this observation window length prevents STEWS scores from being used to monitor newly admitted patients, which limits the utility of STEWS scores.
Comments 3: Here it is certainly of crucial importance to point out in the introduction that the detection of sepsis based on laboratory chemistry or other more complex diagnostics depends primarily on the results and the resulting time delay.
There are considerable difficulties here, even in high-performance medical societies such as the USA.
Response 3: We agree that the time delay involved in laboratory chemistry and other complex diagnostics is an important factor for automated sepsis detection. In the Clinical Variables section in paragraph 2 of page 4, lines 143-146, we list the variables used in our model, none of which involve laboratory chemistry or other more complex diagnostics. We have added the following line in the Clinical Variables section (paragraph 2 of page 4, lines 148-150) to address the lack of laboratory chemistry measurements as a strength of STEWS scores.
The STEWS scores did not use any laboratory values as inputs, which offers an advantage over models that rely on those predictors due to time delays between ordering and releasing laboratory test results.
Comments 4: For what reason were obstetrics (puperal sepsis) or oncology (high-risk sepsis patients) patients excluded?
Response 4: Thank you for your question. Obstetrics and oncology are very different clinical contexts that may benefit from predictors and approaches specific to their respective fields. We sought to focus the study on general non-ICU care settings.
Comments 5: Perhaps the authors also named the SOFA score used so that it is clear that numerous laboratory parameters were included in the identification of sepsis.
Response 5: We did not include SOFA or qSOFA in this study because those are prognostic scores used to stratify patients by risk, while our prediction task was more of a prediction task meant to detect the presence of future sepsis onset. We also did not include numerous laboratory parameters in our sepsis detection score. Rather, other risk scores included for comparison used numerous laboratory parameters.
Comments 6: Did the authors consider that there may have been operationalized blood culture sampling independent of suspected sepsis? (e.g. sterile new insertion of invasive catheters), this could possibly have the consequence that fewer than just under 300 cases of sepsis were observed.
Response 6: We clarify that blood cultures were only considered to be indicative of suspected infection if there was a concurrent initiation of intravenous antibiotics. Blood cultures without initiation of intravenous antibiotics were not considered to indicate suspicion of infection. See the Sepsis Definition section, paragraph 2 of page 3, lines 105-109. To clarify that this reduces the likelihood of false positives in our sepsis definition, we added the following line to the Sepsis Definition section (paragraph 2 of page 3, lines 109-111).
The inclusion of the intravenous antibiotics criterion alongside the blood culture criterion limited the potential for blood culture sampling independent of suspected infection to produce false positives in the gold standard.
Comments 7: Unfortunately, the authors are somewhat neglectful in the classification of their work. Very few hospitals use ML systems at all, and many hospitals, especially in middle or low income countries, have not even implemented any form of digitalization. And even those that do have such a PDMS infrastructure rely to a large extent on manual data entry. In addition to correct data entry, this is one of the sometimes completely unsophisticated implementation problems of all systems. In this respect, the development of STEWS is academically exciting, but how this should help in clinical practice and what steps should be taken to ensure data quality and what clinical consequences should arise from a positive risk score remain unclear.
Response 7: Thank you for your concern. The purpose of this work is to demonstrate how interpretability may be improved without reducing predictive performance in a sepsis prediction risk score. Thus, the help provided to clinical practice will be realized in future work that uses these or similar methods to improve the interpretability and, thus, clinician-friendliness of sepsis prediction scores. To address the concern with timely availability of data to operationalize prediction model implementation, we have added the following lines to the Limitations section (paragraph 2 of page 11, lines 380-383).
Finally, although most hospitals in high-resource countries such as the USA have automated data collection for the predictors used in STEWS scores, other settings with fewer resources still rely on manual data entry. This imposes data availability and timeliness challenges to the implementation of automated scoring systems.
Reviewer 2 Report
Comments and Suggestions for Authors
This study aims to improve the interpretability of an ML-based model for predicting sepsis outside the ICU. The article is well-prepared and the problem definition is detailed. I hope that the following suggestions will contribute to the article.
1. Parameters such as MSE, RMSE, and MAE are examined in regression problems. Why were these parameters not examined for the model? Such parameters determine how much the predicted results differ from the real data.
2. Why are age, and administrative sex important in calculating the STEWS score? Are age and sex-related to sepsis? This relationship seems strange.
3. You do not need to specifically state the doctor in the text. (authors KMK, KM, KK, and GDF) The expression "checked with expert clinicians" is sufficient.
4. What does the second column of Table 1 express?
5. How are PPV and Sensitivity calculated for the regression problem? Please, add the equations.
6. The article is missing in many places in the article. Please verify.
Author Response
Summary
Thank you for taking the time to review this manuscript. Please find the detailed responses below and the corresponding revisions highlighted and in track changes in the re-submitted file.
Point-by-point response to Comments and Suggestions for Authors
Comments 1: This study aims to improve the interpretability of an ML-based model for predicting sepsis outside the ICU. The article is well-prepared and the problem definition is detailed. I hope that the following suggestions will contribute to the article.
- Parameters such as MSE, RMSE, and MAE are examined in regression problems. Why were these parameters not examined for the model? Such parameters determine how much the predicted results differ from the real data.
Response 1: Thank you for bringing this to our attention. Our sepsis prediction task is a classification problem, not a regression problem, so error metrics such as MSE, RMSE, and MAE do not apply. The logistic regression model we used to develop the risk score was engineered to produce a binary prediction for classification, so, despite its name, the model was not used for a regression problem.
Comments 2: 2. Why are age, and administrative sex important in calculating the STEWS score? Are age and sex-related to sepsis? This relationship seems strange.
Response 2: This is an important clarification. In the Clinical Variables section in paragraph 2 of page 4, lines 143-144, we specified that our criteria for selecting clinical variables for calculating the STEWS score were that: (1) the clinical variable was commonly recorded in non-intensive care settings; and (2) the clinical variable was used frequently in previous studies. In addition, older age and sex are known to be risk factors for sepsis. We have added this clarification with the following line in the Clinical Variables section (paragraph 2 of page 4, lines 146-148).
Each of these variables are either direct indicators of patient health or are demographic factors (e.g., age [21] and sex [22]) associated with sepsis risk.
Comments 3: 3. You do not need to specifically state the doctor in the text. (authors KMK, KM, KK, and GDF) The expression "checked with expert clinicians" is sufficient.
Response 3: The manuscript has been updated as suggested. See paragraph 1 of page 6, line 200; and paragraph 6 of page 6, line 246.
Comments 4: 4. What does the second column of Table 1 express?
Response 4: The second column of Table 1 expresses the number and corresponding percentage of patients that belong to a certain category. We added an additional row to Table 1 with subheadings to make it more clear.
Comments 5: 5. How are PPV and Sensitivity calculated for the regression problem? Please, add the equations.
Response 5: We added the following line to the Methods section of the manuscript to clarify PPV and sensitivity calculations. See paragraph 4 of page 2, lines 76-78.
For reference, PPV is defined as the ratio between correctly identified positive cases and all cases identified as positive, and sensitivity is defined as the ratio between correctly identified positive cases and all cases that were actually positive.
Comments 6: 6. The article is missing in many places in the article. Please verify.
Response 6: Thank you for pointing this out. We have checked the manuscript and added missing articles. See for example the Dataset section, paragraph 5 of page 2, line 86; and the Sepsis Definition section, paragraph 1 of page 4, line 131.